# Research Progress of Soil and Vegetation Restoration Technology in Open-Pit Coal Mine: A Review

**Daolong Xu [1], Xiufen Li [2], Jian Chen [1] and Jianghua Li [1,*]**

[1]  National Engineering Laboratory for Cereal Fermentation Technology, Jiangnan University, 1800 Lihu Rd, Wuxi 214122, China
[2]  School of Environment and Civil Engineering, Jiangnan University, Wuxi 214122, China
*   Correspondence: lijianghua@jiangnan.edu.cn; Tel./Fax: +0510-85329031

**Abstract:** Open-pit mining has caused extensive land destruction, including land abandonment and reduction in agricultural land, resulting in serious environmental problems. Ecological restoration and mine reclamation have become important components of the sustainable development strategies in Inner Mongolia, China. Therefore, the rehabilitation of mines and agricultural land is vital and has attracted widespread attention from the Chinese government. In this light, we reviewed the progress of mine restoration technologies in China in recent years and summarized the integrated technology of open-pit mine reclamation with microbial restoration technology as the core, ecological vegetation restoration as the essential, and soil restoration and improvement as the promotion. As a cost-effective and environmentally beneficial technique, combining the microbial recovery technology with vegetation and the recovery of vegetation and the improvement of the soil is widely recommended in the mining reclamation area. At the same time, we comprehensively analyzed the current status and progress of ecological restoration technology and put forward the development direction of green mining in the future. In conclusion, this review can provides guiding the sustainable development of green, ecological mines, as well as provide reference for mining reclamation and agricultural land restoration and other related fields.

**Keywords:** open-pit coal mine; ecological restoration; soil amelioration; vegetation restoration; refuse dumps





## 1. Introduction

Coal resources have always been in a very important position in the structure of nonrenewable energy in the world [1,2]. China is the main coal energy consumer in the word and coal mining has made great contributions to local economic development for years [3]. According to the statistics, seventy-five percent of the added value of global coal production comes from open-pit coal mines. In China, open-pit coal production accounts for 15% of the total coal production [4]. At present, China has gained popularity as the world's largest producer and consumer of coal resources [5,6]. According to forecasts, the total mineable coal reserves in China have a rate of hundreds of millions of tons per year [7]. For example, Inner Mongolia Shengli coalfield is 22.4 billion tons, which has the thickest coal seam and the largest reserves in China. It is also one of the coal fields exceeding 20 billion tons in Inner Mongolia. (Figure 1). The large-scale extraction of coal has put the country's industry in a state of vigorous development and at the same time brings huge economic benefits. However, the long-term and large-scale exploitation of open-pit mines has severely damaged the topography and the natural ecosystem [8,9], such as vegetation degradation [10], soil erosion [11], desertification [12], collapses [13,14], heavy metal pollution [15,16], ground subsidence and air pollution [17]. In addition, coal mining can lead to a reduction in land use and the loss of biodiversity and habitat [18]. It follows that large-scale coal exploitation activities often lead to severe disturbances of the

regional ecology equilibrium [19,20]. After coal mining, the refuse dumps' soil is mostly deep below one hundred meters, and overburdens are devoid of soil characteristics [21], uneven particle size distribution, no soil aggregate structure and poor level of nutrients [22] (Figure 2). Accordingly, open-pit mining operations also change the physical and chemical properties of the soil by markedly destroying soil aggregates and reducing soil fertility [23]. Additionally, the coal gangue and refuse dumps generated by mining occupy a substantial land area [24,25], and not only destroy the geomorphological structure and the landscape, but also lead to the occurrence of soil erosion and geological disasters, including land damage, mudslides, landslides and rocky desertification [26,27]. Furthermore, open-pit mining causes substantial damage to surface vegetation; the direct effect is landscape fragmentation with the loss of grassland and agricultural land [28]. The waste residue and dust from coal mining will also pollute the soil around the mining area, and once the agricultural land is polluted, it is difficult to restore the soil quality by itself. Therefore, the ecological restoration of the mining areas is not only beneficial to the local economic and social stability, but also to the sustainable development of the mining area.

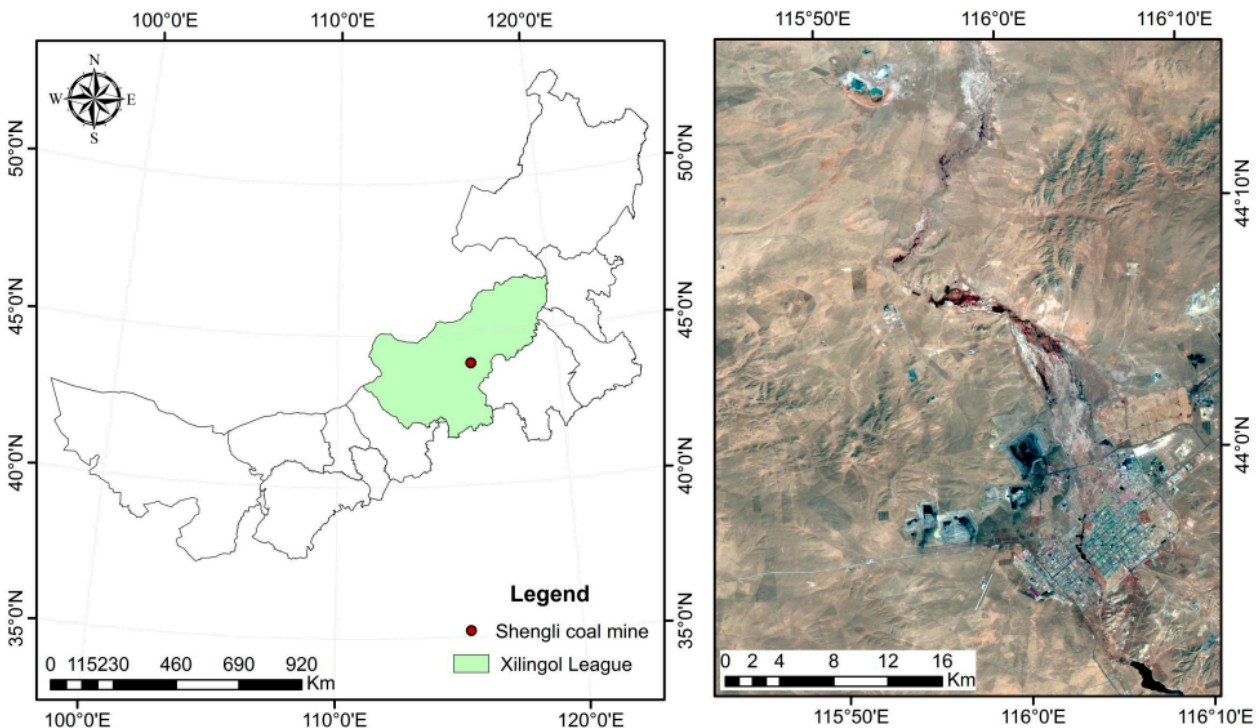

**Figure 1.** Image of the Shengli opencast coal mine in Xilingol League, Inner Mongolia of China.

With the rapid development of modern society, the demand of coal resources is also increasing for people [29]. Massive coal exploitation severely damaged the grassland and its fragile biodiversity. The recovery and management of mines are important and have attracted much focus from the Chinese government. Thus, the improvement of open-pit coal mines has attracted widespread attention. Similarly, another study uncovered that the refuse dumps are considered to be the crucial players of ecological and environmental regression in and around coal mine areas [30]. The soil of the dump site lacks nutrition, and the soil microorganism is relatively rare. In accordance with previous studies, the core environmental issues that reduce plant growth are poor loamy soil of overburden dumps, denuded surfaces and low biomass [31]. Therefore, soil reconstruction is a major component of mine ecological remediation. As previously reported in the literature, we can improve physicochemical properties of the soil, including the soil aggregate stability [32], the moisture content [33], the number of soil layers [34] and the soil configuration [35], so as to improve the effect of vegetation restoration in mining areas [36].

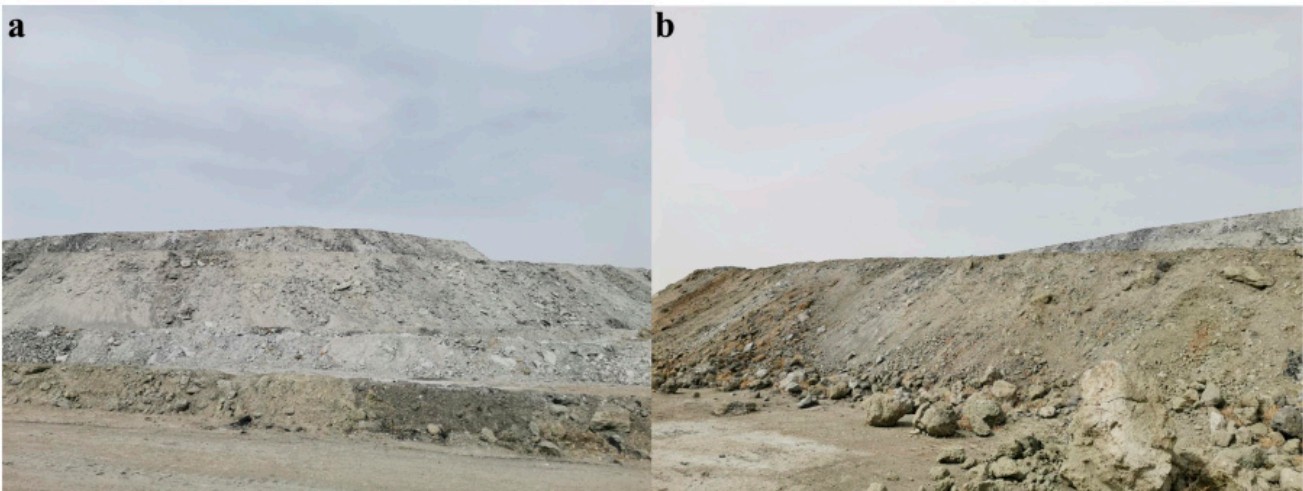

**Figure 2.** A large amount of spoil dumped at the periphery of the open pit after mining. (**a**) below 100 m of the pit. (**b**) soil with poor soil structure and nutrients.

Agricultural land resources are the material basis for human survival. So far, agricultural land has been seriously affected due to mining development. According to statistics, the part of agricultural land overlapping with mining areas in China accounts for about 40% of the country's arable land area. More than 90% of the land destroyed by mining in China's south–north transition zone is agricultural, and most of the coal mining areas in East and North China are in farmland [37,38]. The current reclamation rate of mining areas in China is less than 15%, and the area of land destroyed by mining is still growing rapidly every year, leading to a shortage of land resources. While coal mining brings economic development, it also poses a huge challenge to the quality of agricultural land [39,40]. Therefore, the application of mine ecological restoration technology not only has a positive effect on restoring arable land resources and protecting basic farmland, but also has an important significance in achieving sustainable development of human ecological environment protection.

Currently, many effective studies on vegetation restoration have been carried in mining areas [41,42]. These restoration technologies primarily include physical (change soil, electrodynamics and irrigation) [43], chemical (amendments using soil improver such as biochar, phosphate and limestone) [44], and biological (plant configuration pattern and effective microorganisms) [45]. Biological measures have focused on mycorrhizal symbiosis, and it has great potential to achieve other satisfactory ecological benefits [46]. However, in the actual ecological restoration of mines, a single restoration technology often has certain limitations and cannot achieve the restoration needs. Therefore, in the process of promoting ecological restoration of mining areas, the actual existence of the above complex problems, can not rely on a single method to solve [47].

In the present scenario, we comprehensively analyzed the previous studies, in response to many problems, such as landform destruction, fragile biodiversity, degradation of vegetation diversity, soil erosion, air pollution, and serious desertification caused by mining areas [48,49]. This review aimed to state the main technologies of soil improvement and function enhancement in mining areas, vegetation restoration and reconstruction, and maintaining the sustainable development of the ecosystem (Figure 3). We summarized the feasibility of ecological restoration technologies in mining areas from the perspective of vegetation, soil and microorganisms.

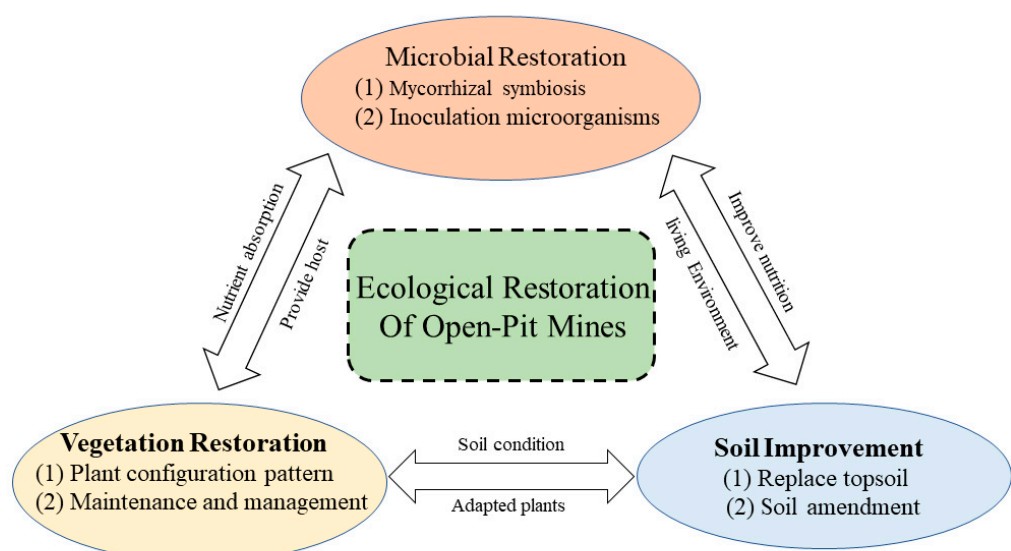

**Figure 3.** Interrelationship of mine ecological restoration technologies.

## 2. Soil Improvement and Function Promotion Technology

Soil is the basis for plant survival, and having fertile soil conditions is the key to vegetation growth and survival [50,51]. Many factors limit plant growth in these soils, including soil pH, toxic substances in the soil, lack of nutrients, etc., creating an environment in which basically no plants can survive. With the increase in national attention to ecological restoration in mining areas, the research on soil improvement technology in mining areas has become the focus of attention of many scholars.

In recent years, a large number of methods and measures have been verified about mine soil restoration, among which rebuilding soil structure and improving fertility by adopting methods such as replacing topsoil and adding amendments to improve and restore soil are the main measures in the ecological restoration process of mine sites at present [52]. As we all know, topsoil is an important part of soil. Therefore, many researchers focus on the improvement of the topsoil. Currently, many scholars believe that topsoil is thin and easily lost or degraded when backfilled or stored, and that the large area of the piled drainage field can no longer be replaced by the original topsoil [53]. Therefore, suitable topsoil needs to be found for replacement. Some scholars in India have used industrial sludge as a topsoil amendment by utilization of industrial waste ETP sludge (effluent treatment plant sludge) to provide a better substrate for the growth of different plant species, thus increasing the soil biomass and improving the soil biomass in a short period of time [30]. Moreover, after 3 years, ETP sludge changes the number of microorganisms: the bacterial population increased from $1.8 \times 10^1$ to $9.8 \times 10^7$ CFU/g, actinomycetes and fungi population increased from nil to $6.5 \times 10^5$ CFU/g and $6.5 \times 10^1$ to $7.5 \times 10^6$ CFU/g, respectively, as depicted in Table S1. Li et al. [54] showed that procuring an air-drying mixture of permafrost and coal gangue instead of expensive topsoil could meet the initial growth requirements of seeds and greatly improved soil nutrients. It has also been shown that tertiary weathered loess and selected unweathered gray sandstone [55], weathered brown sandstone, mixed sandstone and shale as soil substitutes, can all serve to improve soil improvement [56]. Take the example of a coal mine in Inner Mongolia, China. Taking the overlying soil layer of open pit coal mine as the substrate, different additives are added to improve the surface soil of landfill reclamation. The results showed that the optimal formula of the substitute material was a ratio of m (subclay):m (peat) = 20:1 and an applied microbial agent concentration of 0.15 kg/m$^2$ [57]. This method has obvious effect on the improvement of mine surface soil. Lu et al. [58] found that optimum proportion was found to be soil: UCFA: MSLs = 70:20:10 and soil: UCFA: MSLs = 60:20:20; this achieved excellent results in the Baorixile open-pit mine remediation applications in Inner Mongolia,

China (Figure S1). In addition, there are also good technical experiences in the engineering reclamation of soil, such as when piling and discharging soil, creating a slope resembling a pyramid, with the slope <35°, so as to avoid landslides and vegetation seed loss from the site.

Therefore, the reconstruction of topsoil and the addition of improver are important measures of soil improvement and function enhancement technology for coal mine disposal sites, and this method has an important role in promoting the restoration of soil vitality and changing soil microbial diversity in a short period of time for coal disposal sites.

## 3. Vegetation Restoration and Optimal Plant Configuration Pattern Technology

Vegetation restoration is the key to ecological reconstruction in the damaged area of open-pit coal mines, and it is also one of the most common and effective methods to improve the mining environment [59]. Through vegetation restoration and ecological rehabilitation, a stable and efficient artificial vegetation ecosystem is established in the mining area, which provides good ecological environment conditions for the survival of plants and animals [60]. The optimal configuration of vegetation is mainly through the ecological environment conditions of different mining areas, taking local materials according to local conditions, preferably selecting the dominant plants with strong local adaptability and in line with the physiological characteristics of plant growth. The spatial configuration model system of ecological restoration with high stability and sustainability and multiple levels is constructed [61].

At present, according to the research progress of vegetation restoration at home and abroad, vegetation restoration in mining areas should be integrated with artificial restoration means, and then treated by microbial reclamation means to restore the diversity of microorganisms and surface vegetation in mining areas, to screen plants with high tolerance or resistance (pioneer plants) [62,63], and to establish secondary plant communities (companion plants, dominant populations, subdominant populations) on the basis of the growth of pioneer plants, so as to achieve the purpose of ecological restoration. The principles of plant configuration for ecological restoration in mining areas are as follows: (1) the principle of resilience; (2) the principle of ecological adaptability; (3) the principle of plant diversity; (4) the principle of sustained stability of pioneer; (5) the guideline of combining native and exotic plants; (6) the principle of zoning of the site as well as functional rationality. Combined with the current research, the plants used for ecological restoration in mining areas are mainly herbaceous plants. In the process of mine restoration, planting herbs has the advantages of good restoration effect, strong water and soil conservation ability and relatively low investment [64]. On the other hand, we should also consider the law of vegetation development, different planting densities and different planting methods for different plants, and establish various configuration patterns such as shrub monoculture, a mixture of grass and irrigation, and a mixture of grass and seed, etc. The effect of mine vegetation restoration can be improved by selecting native plants with excellent drought resistance and barren resistance.

Based on the review of studies, to achieve vegetation restoration in mining areas, the type of plants has an important impact on the effect of vegetation restoration, so the ratio of different plant species should be considered [65,66]. Table 1 indicates that the ratio of the average abundance value of each plant species in the increased vegetation coverage area was used to determine the proportion of trees, shrubs and herbs to allocate [67]. In general, when selecting species for vegetation configuration, native herbs and shrubs with strong adaptability, fast growth and strong drought resistance should be chosen. It is suggested that trees should be planted properly after the site conditions of vegetation growth are effectively restored or the vegetation community structure tends to be stable.

**Table 1.** Restoration abundance threshold and plant allocation ratio at different site conditions [67].

| Site Conditions | Abundance Value | | Allocation Ratio |
|---|---|---|---|
| | Herb | Shrub | Herbs:Shrub |
| High elevation of sunny-steep slope | 0.1000 | 0.1960 | 1.4:2.8 |
| High elevation of sunny-gentle slope | 0.1188 | 0.2205 | 1.3:2.4 |
| High elevation of sunny-flat slope | 0.1091 | 0.2684 | 1.5:3.6 |
| Low elevation of sunny-steep slope | 0.0949 | 0.2043 | 1.2:2.7 |
| Low elevation of sunny-gentle slope | 0.0921 | 0.2244 | 1.1:2.7 |
| Low elevation of sunny-flat slope | 0.0993 | 0.2320 | 1.4:3.3 |
| High elevation of shady-steep slope | 0.0883 | 0.2417 | 1.9:5.1 |
| High elevation of shady-gentle slope | 0.0829 | 0.2694 | 1.5:4.8 |
| High elevation of shady-flat slope | 0.0900 | 0.2635 | 1.4:4.2 |
| Low elevation of shady-steep slope | 0.0609 | 0.2806 | 1.4:6.4 |
| Low elevation of shady-gentle slope | 0.0719 | 0.2904 | 1.2:5.0 |
| Low elevation of shady-flat slope | 0.0850 | 0.2709 | 1.4:4.5 |

Selecting suitable plant combinations for ecological restoration is an important link in rebuilding stable vegetation communities in open-pit mining areas. According to the previous experience of mine restoration, we concluded that in the process of land reclamation and ecological restoration in mining areas, for slopes with a slope ratio greater than 1:1.7, 3S-OER slope vegetation ecological protection was constructed by a combination of shrub and grass [68], and the mixed grass–bush–tree or bush–tree mode attained the best effect in regulating soil bulk density [69] (Figure 4). These results can improve the basic principles of land reclamation in mining areas and provide a basis for further optimizing land reclamation technology in practice [70]. Additionally, as reported in the previous literature, the vegetation community combination should pay attention to both the horizontal structure of plant diversity and the vertical structure of the combination of trees, shrubs and grasses, and also allocate some species with ornamental characteristics, so that the constructed vegetation system can form a stable vegetation community system and also enhance the landscape function to form a landscape effect with the characteristics of the drainage field landscape [71].

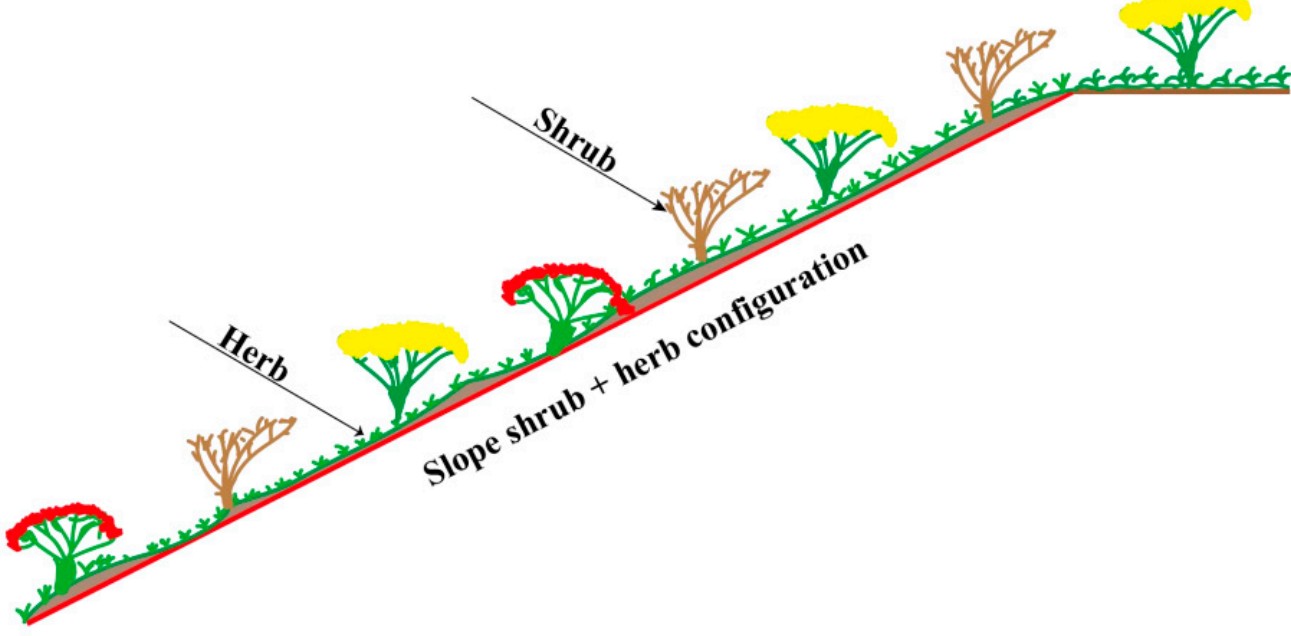

**Figure 4.** The configuration of grass and shrub vegetation on the slope of the dump.

Therefore, the reasonable selection of plant species and configuration patterns used in the ecological restoration process of mining areas, the strict adherence to the development rules of local natural ecosystems, and the scientific establishment of plant configuration patterns to guide the self-recovery process of damaged vegetation systems are also important guidelines for improving the efficiency of vegetation restoration in research mining areas and various mining sites.

## 4. Soil Microbial Restoration Technology

Microbial reclamation technology is an important biological technology for comprehensive soil management and improvement in reclamation areas [72,73]. It takes advantage of the inoculation of microorganisms and can re-establish as well as restore the soil microbial system in the reclaimed area soil that has lost microbial activity by improving plant nutrient conditions and promoting plant growth and development while using the life activities of plant inter-root microorganisms [74]. Microorganisms increase the biological activity of the soil, accelerate the improvement of the soil in the reclamation area, accelerate the transformation process from natural soil to agricultural soil, ripen the raw soil, improve the soil fertility, and thus shorten the reclamation cycle [75,76]. Research has shown that agricultural land contaminated with pesticides and fertilizers can be degraded by microorganisms to reduce the concentration of chemical residues such as pesticides and fertilizers to a manageable level so that they can be used again for agricultural farming [77,78].

In the present study, the study of soil amendment by mycorrhizal fungi has become a hot topic of interest for many scholars [79–81]. Arbuscular mycorrhizal fungi (AMF) are the most widespread and common soil microorganisms in natural soil, and they are a class of endophytic fungi belonging to mycorrhizal fungi [82]. Under the conditions of opencast coal mining with poor soil, symbiotic association can improve nutrient and water uptake efficiency [83], photosynthetic capacity and enzyme activity [84], enhance the root branching capacity and expand the root uptake range, thus promoting plant growth and improving vegetation recovery in mine reclamation areas [85]. It was found that AMF inoculation enhanced the ability of plants to adapt to different types of coal mine spoil complex adversities [86]. This may be due to the fact that in nutrient-deficient coal mine spoil soils, the AMF mycelial structures can help plant roots to extend their nutrient uptake surface area, thus contributing to the enhancement of nutrients in mycorrhizal plants. Bi et al. [87] found that inoculation with AM fungi increased the seed yield and aboveground biomass of wheat by 46.6% and 56.5%, respectively. In opencast coal mining with poor soil, symbiotic associations can improve the efficiency of nutrient and water uptake [88], improve photosynthetic capacity and enzyme activity [89], enhance root branching capacity, and expand the range of root uptake, thereby promoting plant growth and improving vegetation recovery in mine reclamation areas [90]. Song et al. [91] found that inoculation with AMF could significantly improve the biomass of maize and enhance the stress resistance of plants. The inoculation significantly increased the aboveground part by 28.3%, 34.8%, and 24.4%, respectively, while the underground part increased by 37.3%, 20.6%, and 34.8%, respectively (Table 2). In terms of mineral element uptake, a study by Janouskova et al. [92] pointed out that mycorrhizal structures can increase the utilization of nitrogen and phosphorus elements in the soil and enhance the uptake of nitrogen and potassium by plants in mineral soils. In addition, AMF can improve soil structure by reducing the dispersion of soil aggregates and increasing water and nutrient retention in soils, further benefiting plant growth [93].

Previous studies reported that inoculation of biological fertilizers (*Glomus* and *Gigaspora* were separately propagated in a green house in polypropylene) and application of amendments can reduce the toxicity of heavy metals in mines, e.g., chromium, zinc, and cadmium, were significantly reduced to 41%, 43%, and 40%, respectively [94]. Hence, biological fertilizers can be used as an important supporting material for mine remediation. However, mine restoration experiments have most often employed a narrow selection of microbial fertilizer, and examples of effective microbial fertilizers alongside their native

host plants and tracking their co-development are lacking, especially inoculum AMF. Many people have studied that in the process of ecological restoration of mines, inoculation of AMF has a great role in promoting the restoration of plants and soil, which not only improves plant diversity but also improves soil fertility [95,96]. Based on the review of studies, some scholars demonstrate that the co-introduction of native plants and AMF is an effective way to establish species-rich vegetation in post-mining areas. The differences in the plant and AMF variables in response to the addition of seeds or soil inoculum and their interaction are summarized in Table S2 [97]. The table shows that the co-introduction of symbiotic partners resulted in the higher richness, diversity and abundance of plants and AMF than when either partner was introduced individually.

**Table 2.** Effect of inoculation with AM fungi on plant biomass [91]. Notes: the lowercase letters (a, b, c, d) indicate significant differences ($p < 0.05$).

| Item | Mycorrhizal Infection Rate % | Mycelium Density m $g^{-1}$ | Aboveground Biomass g $plant^{-1}$ | Underground Biomass g $plant^{-1}$ | Mycorrhizal Responsiveness % |
|---|---|---|---|---|---|
| Top soil | 0 | 0 | 11.07 ± 1.1 b | 1.58 ± 0.04 b | 29.45 |
| Top soil + M | 83 ± 3 a | 3.61 ± 0.11 a | 83 ± 3 a | 2.17 ± 0.13 a | |
| Sandy soil | 0 | 0 | 8.17 ± 0.27 cd | 0.94 ± 0.13 cd | 25.48 |
| Sandy soil + M | 83 ± 3 a | 3.88 ± 0.09 a | 10.16 ± 0.84 bc | 1.27 ± 0.07 c | |
| Clay soil | 0 | 0 | 7.08 ± 0.59 d | 0.7 ± 0.06 d | 11.85 |
| Clay soil + M | 80 ± 2 a | 1.54 ± 0.21 b | 7.86 ± 0.34 bcd | 0.82 ± 0.05 c | |
| S_C soil | 0 | 0 | 10.57 ± 0.45 b | 1.61 ± 0.07 b | 34.83 |
| S_C soil + M | 83 ± 3 a | 3.79 ± 0.13 a | 14.25 ± 0.8 a | 2.17 ± 0.14 a | |

Therefore, the application of microbial technology to mine land reclamation has achieved better ecological effects. The development and use of some microbial agents for mine remediation has also become a research hotspot in this field [98,99]. These microbial products are non-toxic, environment-friendly, and act as potential tools for plant growth promotion and resilience. However, the microbial remediation conditions of mine ecology are harsh, the remediation time is long, the specific microorganisms are generally effective only for a single pollutant, and the selection of microorganisms is easily restricted by environmental conditions, which is not suitable for large-scale damage management. Accordingly, the microbial remediation technology needs further comprehensive and in-depth research; especially the pilot test of the microbial bacterial agent products and their industrialization still has a large limitation.

## 5. Efficient Management and Monitoring Technology

In recent years, the development of mine ecological restoration technology in China has gradually matured. Moreover, the level of post-remediation management and monitoring has been greatly improved. Effective management measures and timely feedback on the effectiveness of mine reclamation are also an important part of ecological restoration. This includes a series of management monitoring of plant growth, soil nutrition and irrigation [100]. In terms of vegetation management measures, to prevent the loss of sown grass seeds, a layer of grass thatch or fabric-resembling material is covered to protect the seeds from being blown away by the north wind and also to increase the soil temperature and improve seed germination and vegetation cover [101]. For example, we conducted relevant experiments in the open-pit dump of the Shengli Coal Mine in Inner Mongolia, where the slope was covered with straw, which significantly increased the seed germination rate (Figure 5).

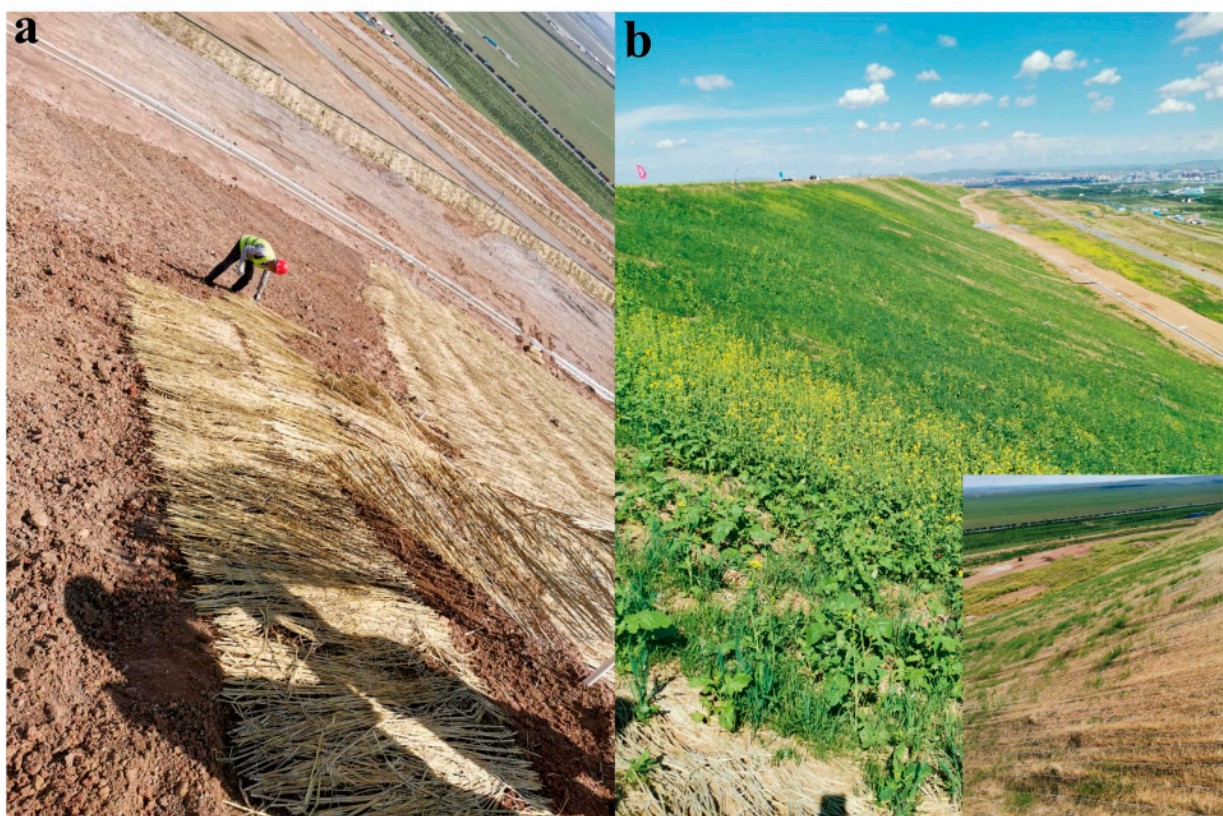

**Figure 5.** Application of seeded spoil dumps covered with fibrous straw mulch in revegetation. (**a**) Covering of the sowed with straw thatch. (**b**) Comparison of refuse dumps before and after restoration.

At present, China's post-mine ecological restoration results monitoring technology is still in the development stage. The most common way of environmental monitoring is still a relatively primitive field survey and satellite remote sensing images [102]. These methods are not only inefficient and costly, but also the timeliness and accuracy of the obtained data need to be improved [103]. Therefore, efficient and accurate detection technology plays an important role in improving the mine restoration effect. With the development of UAV technology, the advantages of its application in the process of mine rehabilitation are also reflected. Here, in Table 3, we also summarize and compare the advantages and disadvantages of manual surveying, satellite remote sensing and unmanned aerial vehicles' remote sensing data collection based on previous research [104,105].

Accordingly, some emerging technologies such as satellite remote sensing and unmanned aerial vehicles have been widely used. For example, they have been used for ecological restoration monitoring, basic soil conditions in the target area monitoring, infrastructure layout monitoring, and plant growth conditions monitoring, etc. Wang et al. [106] used remote sensing data to monitor and evaluate vegetation restoration. Park and Choi [107] discussed the analysis of mine ecological restoration using data obtained from unmanned aerial vehicles. It has been proved that the technology not only collects data efficiently but also ensures real-time data. In the future, China's management and monitoring technology in mine ecological restoration applications will be further deepened, and the focus of research will mainly include dynamic monitoring, comprehensive management and benefit assessment [108]. Through this technical means, we can give full play to the advantages of supervision, develop a reasonable restoration plan, and finally realize the optimal benefit assessment of mine reclamation.

**Table 3.** A comparison of field investigation, satellite remote sensing, and unmanned aerial vehicle remote sensing technologies.

| Evaluation Indicators | Field Investigation | Satellite Remote Sensing | Unmanned Aerial Vehicles Remote Sensing |
|---|---|---|---|
| Cost budget | High, surveying and mapping costs account for 15% of the total cost | Moderate, requires a higher cost to complete | Low operating cost and can be used multiple times |
| Work efficiency | Inefficient and time-consuming | General efficiency, has a time lag in regional monitoring. | Relatively high efficiency simple and fast operation |
| Accuracy | Lower, there will be human error | Generally Influenced by many external factors | Higher, collect information efficiently |
| Data aging | Low, long periodic table | Generally, longer cycle | High, timely and accurate data |
| Convenient and practical | Inconvenient, the terrain is complex and it is difficult to collect information | Better, convenient and practical | Better, convenient and practical |

## 6. Outlook

Ecological restoration of mines is a long-term and complex process and a worldwide challenge. Therefore, mine vegetation restoration and soil improvement are important research directions for future ecological construction and ecological protectors. This paper describes the technology of soil improvement and function enhancement based on the regulation mechanism of plant–microbial–soil interactions and the technical system of vegetation–soil feedback interactions monitoring and management and proposes the application prospect of soil restoration and improvement technology, vegetation community diversity reconstruction technology and microbial restoration technology in solving the ecological restoration of mines. Moreover, future research should integrate multi-level and multi-faceted studies on geology, hydrology, soil, vegetation, microorganisms, animals, and climate to explore mine remediation technology and the molecular mechanisms involved in it more comprehensively and deeply.

At the same time, combined with the current status and progress of mine ecological restoration technology, we propose some research and development directions for future mine restoration to lay the foundation for green and sustainable development of future mine development.

(1) Microbial remediation technology, as a green and efficient remediation pathway, has a broad prospect in the field of mine remediation. However, further research is needed on the acquisition of microbial strains and the tolerance of microorganisms to the environment. For example, some new autotrophic microbial strains can be obtained from factory sludge, domestic waste and solid waste through composting and fermentation technology. Microbial tolerance can be enhanced by artificial domestication or genetic modification techniques to enhance remediation efficiency.

(2) As an important indicator of mine remediation, the core of soil remediation is soil fertilization, improvement and maturation. The future application of soil organic fertilizer can effectively improve soil quality and can establish a mature soil–plant–microorganism biological system. However, we need to do further research on the mechanism and influencing factors of organic fertilizer for mine soil remediation in the future.

(3) Phytoremediation technology occupies an important position in the field of mine ecological restoration as an effective restoration pathway for green ecology and environmental protection. However, phytoremediation faces problems, such as long restoration periods and incomplete restoration, that still need to be solved. Future phytoremediation research can modify plant genes by genetic means to enhance plant tolerance and restoration ability (such as absorption and transformation), thus shortening the restoration cycle and improving restoration efficiency.

(4) Finally, our comprehensive evaluation methods of restoration effects still need further research. For example, in the future, we can combine some high-tech methods such as high-light remote sensing and AI to make a comprehensive evaluation of soil–soil characteristics and plant growth characteristics.

**Supplementary Materials:** The following supporting information can be downloaded at: https://www.mdpi.com/article/10.3390/agriculture13020226/s1, Figure S1: Requirements for plant growth experiments and determination of optimal scenarios [58]; Table S1: Microbial characteristics of the coal mine spoil [30]; Table S2: Differences in plant and arbuscular mycorrhizal fungal (AMF) variables in response to the addition of seeds or soil inoculum and their interaction. Degrees of freedom (df), F-statistics and p values are reported. Significance at 0.001 ***, 0.01 ** and 0.05 * levels shown. Ns—not significant. The results of generalized linear models with the addition of seeds and soil inoculum as fixed effects are shown [97].

**Author Contributions:** D.X.: investigation, methodology, writing—original draft, writing—review, and editing. X.L.: formal analysis. J.L. and J.C.: supervision, resources, writing—review, and editing. All authors have read and agreed to the published version of the manuscript.

**Funding:** This research received no external funding.

**Institutional Review Board Statement:** Not applicable.

**Informed Consent Statement:** Not applicable.

**Data Availability Statement:** Not applicable.

**Conflicts of Interest:** The authors declare no conflict of interest.

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
