# Peer review of "Research Progress of Soil and Vegetation Restoration Technology in Open-Pit Coal Mine: A Review"

_agriculture, doi:10.3390/agriculture13020226_

Round 1

Reviewer 1 Report

Topic of the paper highly relevant. Only corrections are required.

Overall paper is good and presented the valuable information but few questions;
1. How many paper you collected and how many you consider for the study.
2. mention the data/paper collection process in methodology section along with data analysis
3. Conclusion and recommendation add in the paper
4. Rest is okay

Author Response

We have highlighted the changes in purple according to reviewer #1 comments in the annotated version of the revised manuscript (Revision, changes marked).

1.How many paper you collected and how many you consider for the study.

We appreciate the reviewer’s insightful comments.We have a total of 108 references for the study(Lines 387-731).

2.mention the data/paper collection process in methodology section along with data analysis

We are thankful to the reviewer’s comments. We have carefully checked and corrected this mistake in the revised manuscript.

Lines 157-160 : Added “In addition, there are also good technical experiences in the engineering reclamation of soil, such as: when piling and discharging soil, creating a slope resembling a pyramid, with the slope < 35°, so as to avoid landslides and vegetation seed loss from the site.”

Lines 217-223 : Added “Also, as reported in the previous literature, the vegetation community combination should pay attention to both the horizontal structure of plant diversity and the vertical structure of the combination of trees, shrubs and grasses, and also allocate some species with ornamental characteristics, so that the constructed vegetation system can form a stable vegetation community system and also enhance the landscape function to form a landscape effect with the characteristics of the drainage field landscape [71].”

Lines 280-287 : Added “Many people have studied that in the process of ecological restoration of mines, inoculation of AMF has a great role in promoting the restoration of plants and soil, which not only improves plant diversity but also improves soil fertility [95,96]. Based on the review of studies, some scholars demonstrate that co-introduction of native plants and AMF is an effective way to establish species rich vegetation in post-mining areas, the differences in plant and AMF variables in response to addition of seeds or soil inoculum and their interaction have been summarized in Table S2[97]. ”

3.Conclusion and recommendation add in the paper

We appreciate the reviewer’s insightful comments. we have added conclusion and recommendation in the revised manuscript.

Lines 339-351: Added “Ecological restoration of mines is a long-term and complex process, and a worldwide challenge. Therefore, mine vegetation restoration and soil improvement are important research directions for future ecological construction and ecological protectors. This paper describes the technology of soil improvement and function enhancement based on the regulation mechanism of plant-microbial-soil interactions and the technical system of vegetation-soil feedback interactions monitoring and management, and proposes the application prospect of soil restoration and improvement technology, vegetation community diversity reconstruction technology and microbial restoration technology in solving the ecological restoration of mines. Also, future research should integrate multi-level and multi-faceted studies on geology, hydrology, soil, vegetation, microorganisms, animals, and climate to explore mine remediation technology and the molecular mechanisms involved in it more comprehensively and deeply”.

Lines 352-379: Added “At the same time, combined with the current status and progress of mine ecological restoration technology, we propose some research and development directions for future mine restoration to lay the foundation for green and sustainable development of future mine development.

(1) Microbial remediation technology, as a green and efficient remediation pathway, has a broad prospect in the field of mine remediation. However, further research is needed on the acquisition of microbial strains and the tolerance of microorganisms to the environment. For example, some new autotrophic microbial strains can be obtained from factory sludge, domestic waste and solid waste through composting and fermentation technology. For microbial tolerance can be enhanced by artificial domestication or genetic modification techniques to enhance remediation efficiency.

(2) As an important indicator of mine remediation, the core of soil remediation is soil fertilization, improvement and maturation. The future application of soil organic fertilizer can effectively improve soil quality and can establish a mature soil-plant-microorganism biological system. However, we need to do further research on the mechanism and influencing factors of organic fertilizer for mine soil remediation in the future.

(3) Phytoremediation technology occupies an important position in the field of mine ecological restoration as an effective restoration pathway for green ecology and environmental protection. However, phytoremediation faces problems such as long restoration period and incomplete restoration still need to be solved. Future phytoremediation research can modify plant genes by genetic means to enhance plant tolerance and restoration ability (such as absorption and transformation), thus shortening the restoration cycle and improving restoration efficiency.

(4) Finally, our comprehensive evaluation methods of restoration effects still need further research. For example, in the future, we can combine some high-tech methods such as high-light remote sensing and AI to make a comprehensive evaluation of soil-soil characteristics and plant growth characteristics”.

4.Rest is okay

We are thankful to the reviewer’s comments. We have carefully checked and corrected this mistake in the revised manuscript.

Reviewer 2 Report

As an overview, it is too cursory and only signals some solutions. Many literature items are used forcefully and confirm obvious facts, while elsewhere literature sources are completely omitted.

It was pointed out that the review concerns Chinese Open-Pit Coal Mines, while the paper discusses solutions from different angles, without detailed recognition of Shengli opencast coal mine in Xilingol League, which was mentioned in the introduction.

In my opinion, there is a lack of materials and methods - a detailed recognition of the location (one or many) and a description of the solutions used, and not just a hint about them. Maybe it will be easier to change the title and remove the section on selected bets while discussing others?

There are some misspellings (eg. line 68 Germanys).

Paragraph from line 87 should be moved to site characteristic.

Not all references are crucial for this article, eg. 34, 35 - they were used only once to confirm obvious statements. Some other statements were left without references (eg. line 138: "Some researchers in India").

The article should focus on "Chinese Open-Pit Coal Mines" it's unclear whether it does or describe other open-pit mines across the world.

Fig. 6 - what is shown on OX and OY?

Titles in line 230 and 295 should be change

Table 3 - based on what?

Author Response

Revision Notes

Dear Editor:

Thank you for your kind letter of “Agriculture-2126182 - Editor decision -Major Revisions” on December 28, 2022.

Generally, we appreciate the editor and reviewer’s insightful comments, which are helpful for improving the manuscript. We are thankful to reviewer  #1、#2、#3 and #4 comments , which are all valuable and very helpful for revising and improving our paper. Based on editor and reviewers' comments and requests, we have made moderate modification in the revised manuscript. Below is a summary of our responds to the reviewers' comments.

In revision notes, the line numbers refer to the PDF vision of the revised manuscript.

We have highlighted the changes in red according to reviewer #2 comments in the annotated version of the revised manuscript (Revision, changes marked).

1.As an overview, it is too cursory and only signals some solutions. Many literature items are used forcefully and confirm obvious facts, while elsewhere literature sources are completely omitted.

We are very sorry for not clearly describing this content in the previous manuscript. We have added some informative and key sentences and literature in the revised manuscript.

Lines 39-48: Added the sentence of “According to the statistics, seventy-five percent of the added value of global coal production comes from open-pit coal mine, in China , open-pit coal production accounts for 15% of the total coal production [4]. At present, China has gained popularity as the world’s largest producer and consumer of coal resources [5,6]. According to forecasts, the total mineable coal reserves in China with a rate of hundreds of millions of tons per year[7]. Such as, the Shengli coalfield contains 22.4 billion tons of coal, which is the lignite coalfield with the thickest coal seam and the largest reserves in China, and is also one of the three coalfields in Inner Mongolia Autonomous Region with more than 20 billion tons, and has been included in the national large-scale coal power base”.

Lines 532-536: Added the literature “[50] Zhu S C, Zheng H X, Liu W S, et al. Plant-Soil Feedbacks for the Restoration of Degraded Mine Lands: A Review. Frontiers in Microbiology, 2022, 12: 4238-4238. and [51] Nadalia D and Pulunggono H B. Soil characteristics of post-mining reclamation land and natural soil without top soil. Journal of Degraded and Mining Lands Management, 2020(2).”

Lines 120: Changed “Technologies for soil improvement and promotion of function” to “Soil improvement and function promotion technology”.

Lines 166:Changed “Technologies for vegetation restoration and optimal plant configuration pattern ” to “Vegetation restoration and optimal plant configuration pattern technolog”.

Lines 230:Changed “Technologies for soil microbial restoration technology” to “Soil microbial restoration technology”.

Lines 301:Changed “Technologies for efficient management and monitoring ecological” to “Efficient management and monitoring technology”.

2.It was pointed out that the review concerns Chinese Open-Pit Coal Mines, while the paper discusses solutions from different angles, without detailed recognition of Shengli opencast coal mine in Xilingol League, which was mentioned in the introduction.

We appreciate the reviewer’s insightful comments.We have deleted some redundant sentences in the revised manuscript.

.

Lines 87-97: Deleted “The Shengli opencast coal mine in Xilingol League, Inner Mongolia of China was selected as an example to research restoration vegetation and it was an Open-Pit Coal Mine, as shown in (Fig 1). The Shengli coalfield contains 22.4 billion tons of coal, which is the lignite coalfield with the thickest coal seam and the largest reserves in China, and is also one of the three coalfields in Inner Mongolia Autonomous Region with more than 20 billion tons, and has been included in the national large-scale coal power base (Fig 2). The coal type of Shengli coal field is lignite, with 1.890 billion tons of recoverable reserves in the well field, with reliable resources and superior mining conditions. The soil type of coal mine mainly consists of chestnut calcium soil, meadow chestnut calcium soil, meadow soil, etc. Due to the degradation of the meadow, sandy and gravelly chestnut calcium soil has been formed, with low soil organic matter content and poor soil fertility (Fig 3).”

  1. In my opinion, there is a lack of materials and methods - a detailed recognition of the location (one or many) and a description of the solutions used, and not just a hint about them. Maybe it will be easier to change the title and remove the section on selected bets while discussing others?

We appreciate the reviewer’s insightful comments, and sorry for our negligence. We have change the title and remove the section on selected bets.

Lines 64-76: We have deleted “Due to the leading industrialization and environmental protection concept, the foreign research related to mine remediation was started earlier and the concept was advanced and perfect [21]. As reported in the previous literature, the reclamation of post-mining areas has just reached 100 years from the first applications. In the 1920s, the post-mining areas began to be reclaimed after lignite mining in Germanys [22]. Since the 1950s, a number of restoration technologies have been developed on the European continent, including these from Germany, Australia, France and Poland [23,24,25]. The basis of the early activities was exactly ecological reclamation. Then the so-called a biodynamic method has been introduced, supported by different cultivation activities. Finally, from the 1980s, the search for other reclamation technologies began, for example in the USA there was a strong interest in phytoremediation [26]. While conventional technologies have been applied for more than 100 years and are fit for purpose, the current paradigm shift in open mining repair model is focus on soil microorganism technology.”

Lines 120: Changed “Technologies for soil improvement and promotion of function” to “Soil improvement and function promotion technology”.

Lines 166:Changed “Technologies for vegetation restoration and optimal plant configuration pattern ” to “Vegetation restoration and optimal plant configuration pattern technolog”.

Lines 230:Changed “Technologies for soil microbial restoration technology” to “Soil microbial restoration technology”.

Lines 301:Changed “Technologies for efficient management and monitoring ecological” to “Efficient management and monitoring technology”.

4.There are some misspellings (eg. line 68 Germanys).

We appreciate the reviewer’s insightful comments, and sorry for our negligence. We have deleted the misspellings sentences in the revised manuscript.

Line 68: Deleted “ In the 1920s, the post- 67 mining areas began to be reclaimed after lignite mining in Germanys [22]”.

5.Paragraph from line 87 should be moved to site characteristic.

We appreciate the reviewer’s insightful comments, and sorry for our negligence. Considering the overall structure of the article, we have deleted this part.

Lines 87-97: Deleted “The Shengli opencast coal mine in Xilingol League, Inner Mongolia of China was selected as an example to research restoration vegetation and it was an Open-Pit Coal Mine, as shown in (Fig 1). The Shengli coalfield contains 22.4 billion tons of coal, which is the lignite coalfield with the thickest coal seam and the largest reserves in China, and is also one of the three coalfields in Inner Mongolia Autonomous Region with more than 20 billion tons, and has been included in the national large-scale coal power base (Fig 2). The coal type of Shengli coal field is lignite, with 1.890 billion tons of recoverable reserves in the well field, with reliable resources and superior mining conditions. The soil type of coal mine mainly consists of chestnut calcium soil, meadow chestnut calcium soil, meadow soil, etc. Due to the degradation of the meadow, sandy and gravelly chestnut calcium soil has been formed, with low soil organic matter content and poor soil fertility (Fig 3).”

6.Not all references are crucial for this article, eg. 34, 35 - they were used only once to confirm obvious statements. Some other statements were left without references (eg. line 138: "Some researchers in India").

We appreciate the reviewer’s insightful comments, and sorry for our negligence. We have replaced some references and added some references.

Lines 122: Changed “[34]Du, S.H.; Xiong, Z.Q.; Wang,Y.C.; Guo, L. Quantifying the multilevel effects of landscape composition and configuration on land surface temperature. Remote Sensing of Environment. 2016, 178: 84-92. and [35]Levi, N.; Hillel, N.; Zaady, E.; Rotem, G.; Ziv, Y.; Karneli, A.; Paz-Kagan,T. Soil quality index for assessing phosphate mining restoration in a hyper-arid environment-Science Direct. Ecological Indicators. 2021, 125, (107571).

” to “[50] Zhu S C, Zheng H X, Liu W S, et al. Plant-Soil Feedbacks for the Restoration of Degraded Mine Lands: A Review. Frontiers in Microbiology, 2022, 12: 4238-4238. and [51] Nadalia D and Pulunggono H B. Soil characteristics of post-mining reclamation land and natural soil without top soil. Journal of Degraded and Mining Lands Management, 2020(2)”.

Lines 139: Added “[30] Jambhulkar H P and Hemlata P. Eco-restoration approach for mine spoil overburden dump through biotechnological route. Environmental monitoring and assessment. 2019, 191(12): 1-16.”.

7.The article should focus on "Chinese Open-Pit Coal Mines" it's unclear whether it does or describe other open-pit mines across the world.

We are thankful to the reviewer’s comments, and strongly cherish this opportunity. Therefore, we attach great importance to the above comments. We have added some informative focus on "Chinese Open-Pit Coal Mines".

Lines 39-50: Changed “Seventy-five percent of the economic value of coal production worldwide comes from open pit coal mining, and in China, open pit coal mine production occupies 15% of the total coal production [2,3]. In 2006, the USA, Russia, India, China, Australia and South Africa, produced 81.9% of the total coal extracted throughout the world [4,5]. Since the industrial revolution of the 18th and 19th centuries, coal mining in the past more than 200 years has produced great contribution to the national economy, it has inevitably bring serious environmental and land problems [6,7].” to “According to the statistics, seventy-five percent of the added value of global coal production comes from open-pit coal mine, in China , open-pit coal production accounts for 15% of the total coal production [4]. At present, China has gained popularity as the world’s largest producer and consumer of coal resources [5,6]. According to forecasts, the total mineable coal reserves in China with a rate of hundreds of millions of tons per year [7]. Such as, the Shengli coalfield contains 22.4 billion tons of coal, which is the lignite coalfield with the thickest coal seam and the largest reserves in China, and is also one of the three coalfields in Inner Mongolia Autonomous Region with more than 20 billion tons, and has been included in the national large-scale coal power base (Fig 1).The large-scale extraction of coal has put country’s industry in a state of vigorous development and at the same time brings huge economic benefits”.

8.Fig. 6 - what is shown on OX and OY?

We appreciate the reviewer’s insightful comments, and sorry for our negligence. We have deleted some redundant figure in the revised manuscript.

Lines 204: Deleted “Liu et al.[29] investigated the calculated proportions of different plant distributions in the 12 site conditions in the semi-arid mining areas of western China, with higher proportions of herbs and shrubs than trees. Herbaceous plants and shrubs play an important role in the revegetation of semi-arid mining areas (Fig. 6).” 

9.Titles in line 230 and 295 should be change.

We appreciate the reviewer’s insightful comments, and sorry for our negligence. We have change titles in line 230 and 295.

Lines 230:Changed “Technologies for soil microbial restoration technology” to “Soil microbial restoration technology”.

Lines 301:Changed “Technologies for efficient management and monitoring ecological” to “Efficient management and monitoring technology”.

10.Table 3 - based on what?

We appreciate the reviewer’s insightful comments, and sorry for our negligence. We have added some references to support table 3.

Lines 324: Added references“ [104]Rasmussen J, Azim S, Jensen S M, et al. The challenge of reproducing remote sensing data from satellites and unmanned aerial vehicles (UAVs) in the context of management zones and precision agriculture. Precision Agriculture, 2020. and [105]Liao X, Yue H, Liu R, et al. Launching an unmanned aerial vehicle remote sensing data carrier:concept,key components and prospects. International Journal of Digital Earth, 2020(010):013.”

Thank you again for your comments. We hope we could learn more from you. Finally, we are very grateful to the reviewer’s and editor’s understanding and affirmation again. We wish the article can be published in Agriculture.

Reviewer 3 Report

As the main goal, the manuscript proclaims summarizing the integrated technologies applied to the open pit mine reclamation with microbial restoration technology as the core, ecological vegetation restoration as the essential, and soil improvement as the promotion, which is an actual task nowadays. However, the presented review materials, observed investigations, and references collected are not permitted authors to fully achieve this ambitious goal.

The full number of articles included in the reference list is 82 which is not enough for a comprehensive review. The authors did not explain which methodology they utilized. How did they provide the selection of the articles? Did they overview any scientific database like Web of Knowledge, Scopus, etc.? Or they just analyzed the known before publications and their own research?     

The presented in the introduction statistics, data – where they have been taken?

The authors presented the general data about open coal mining as for 2006, would be good to introduce more recent data. The introduction of the state with the revitalization of post-mining sites across the world is rather fragmentary. Either authors have to enrich the state with revitalization with exact cases across the world or to concentrate only at the situation in China. Currently, the global state is presented very fragmentary and does not show the real state of the art, a number of countries with intensive coal mining: GB, Czech Republic, etc. are missing.   

The motivation why the Shengli opencast coal mine in Xilingol League, Inner Mongolia of China was selected as one case study, is missing. Moreover, in the following investigation authors did not refer to this particular case and it became unclear whether the reviewed cases (rather often without appropriated citations) are connected with this particular site? 

Figure 2 and Figure 3 are not connected with the main goal of the review which pretends to overview existing in China technologies. The pictures are not much informative, and it is not clear what was the reason to include them in the manuscript.

Rows 120-125 present common general knowledge, well known from the Class Books as per open-pit mining reclamation, but they are not informative.

Table S2 of the current manuscript is very similar to Table 2 of the reference [30]: even the titles of both tables are the same and conclusions from them.  It is not explained how Table S2 was created, and to which extent it is original.

Row 137:  ETP sludge – the full title of the meaning, not abbreviation.

Conclusions are rather trivial and weak: the well-known methods used for ecological restoration of post-mining sides are calculated: soil fertilization, phytoremediation, microbial remediation and their common influence.  Which new knowledge has been created as a result of the study?  

More comprehensive research was done before with good conclusions, like in the reference: Feng, H.B.; Zhou, J.W.; Zhou, A.G.; Bai, G.Y.; Li, Z.X.; Chen, H.N.; Su, D.H.; Han, X. Grassland Ecological Restoration Based on the Relationship between Vegetation and Its Below-ground Habitat Analysis in Steppe Coal Mine Area. Science of The Total Environment. 2021, 78: 146221.  What was the reason to present the conclusions of this reference again in the reviewed manuscript?

 In many places the respected citations are missing, for example:   

1.       Row 92:   The Shengli coal fields… ‘has been included in the national large-scale coal power base” however, the reference is to figure 2, not to the database.

2.       Rows 133-134. It is written ‘some researchers in India”: references are missing.

3.       Rows 193-194. It is written “Based on a review of published studies, to achieve vegetation restoration in mining areas, the plant type has an important impact on the effect of vegetation restoration; thus, the ratio of different plant species should be considered”: references are missing.

4.       Rows 213-215. It is written “Based on our previous experience in mine restoration, we concluded that the mixed grass-bush-tree or bush-tree mode attains the best effect in regulating soil bulk density”: references are missing.  

5.       Rows 277-278. It is written “In addition, some studies demonstrated that co-introduction of native plants and AMF is an effective approach to establish species-rich vegetation in post-mining areas”: references are missing. 

6.       Rows 309-313. It is written “Currently, technologies for post-mine ecological restoration monitoring in China are still in the development stage. The most common method of environmental monitoring is field surveys and satellite remote sensing, which are relatively primitive. These methods are not only inefficient and costly but the timeliness and accuracy of the obtained data need to be improved”: references are missing.  

7.       Rows 315-317. It is written, “In Table 3, we summarize and compare the advantages and disadvantages of manual surveying, satellite, remote sensing, and UAV remote sensing data collection based on previous research”: which research? references are missing.

The review has to be significantly revised, the missing research methods and methodology have to be presented, and a more comprehensive analysis has to be done. English has to be improved essentially, there are some grammar mistakes, and the layout of the sentences has to be improved. 

Author Response

Revision Notes

Dear Editor:

Thank you for your kind letter of “Agriculture-2126182 - Editor decision -Major Revisions” on December 28, 2022.

Generally, we appreciate the editor and reviewer’s insightful comments, which are helpful for improving the manuscript. We are thankful to reviewer  #1、#2、#3 and #4 comments , which are all valuable and very helpful for revising and improving our paper. Based on editor and reviewers' comments and requests, we have made moderate modification in the revised manuscript. Below is a summary of our responds to the reviewers' comments.

In revision notes, the line numbers refer to the PDF vision of the revised manuscript.

We have highlighted the changes in yellow according to reviewer #3 comments in the annotated version of the revised manuscript (Revision, changes marked).

1.As the main goal, the manuscript proclaims summarizing the integrated technologies applied to the open pit mine reclamation with microbial restoration technology as the core, ecological vegetation restoration as the essential, and soil improvement as the promotion, which is an actual task nowadays. However, the presented review materials, observed investigations, and references collected are not permitted authors to fully achieve this ambitious goal.

We are very sorry for not clearly describing this content in the previous manuscript. We have added some informative and key sentences and literature in the revised manuscript.

Line: 36-37: Added: “Coal resources has always been in a very important position in the structure of nonrenewable energy in the world [1,2].”

Lines: 50-59:Added: “However, the long-term and large scale exploitation of open-pit mines has severely damaged the topography and the natural ecosystem [8,9],such as vegetation degradation [10], soil erosion [11], desertification [12], collapses [13,14], heavy metal pollution [15,16], ground subsidence and air pollution [17]. In addition, coal mining can lead to a reduction in land use and the loss of biodiversity and habitat [18]. It follows that large-scale coal exploitation activities often lead to severe disturbances of regional ecology equilibrium [19,20]. After coal mining, the refuse dumps soil is mostly deep below one hundred meters, and overburdens are devoid of soil characteristics [21], uneven particle size distribution, no soil aggregate structure and poor level of nutrients [22].” 

Line: 709-712: Added: “[102]Shuai, S.; Zhang, Z.; Lyu, X.; et al. Remote sensing monitoring of vegetation phenological characteristics and vegetation health status in mine restoration areas. Nongye Gongcheng Xuebao/Transactions of the Chinese Society of Agricultural Engineering, 2021, 37(4):224-234.”

Line: 713-716: Added: “[103]Qi, Y.N.; Liao, S.B.; Wang, Q.L.; Research on evaluation of mining area ecological security based on GF-1 satellite imagery-taking Fushun West open-pit mine for example. IOP Conference Series: Earth and Environmental Science, 2021, 783(1):012128 (7pp).”

Line: 717-720: Added: “[104]Rasmussen, J.; Azim, S.; Jensen, S.M.; et al. The challenge of reproducing remote sensing data from satellites and unmanned aerial vehicles (UAVs) in the context of management zones and precision agriculture. Precision Agriculture, 2020.”

Line: 721-723: Added: “[105]Liao, X.; Yue, H.; Liu, R.; et al. Launching an unmanned aerial vehicle remote sensing data carrier:concept,key components and prospects. International Journal of Digital Earth, 2020(010):013.”

Line: 724-725: Added: “[106] Park, S and Choi, Y. Applications of Unmanned Aerial Vehicles in Mining from Exploration to Reclamation: A Review. Minerals. 2020, 10(8): 663.

Line: 725-726: Added: “[107] Han, Z.Y and Han. Research progress in the application of Unmanned Aerial Vehicles technology in mine restoration projects. Henan Science an Technology. 2021,40(21): 62-65.”

2.The full number of articles included in the reference list is 82 which is not enough for a comprehensive review. The authors did not explain which methodology they utilized. How did they provide the selection of the articles? Did they overview any scientific database like Web of Knowledge, Scopus, etc.? Or they just analyzed the known before publications and their own research?

We appreciate the reviewer’s insightful comments. We have added some informative and key sentences and literature in the revised manuscript.

Line: 387-390: Added: “[1] Shi, J.X.; Huang, W.P.; Han, H.J.; Xu. C.Y. Pollution control of wastewater from the coal chemical industry in China: Environmental management policy and technical standards. Renewable and Sustainable Energy Reviews. 2021, 143(4):110883.”

Line: 391-393: Added: “[2] Wu, Z. H.; Lei, S.G.; Lu, Q. Q.; Zheng, F.B. Impacts of Large-Scale Open-Pit Coal Base on the Landscape Ecological Health of Semi-Arid Grasslands. Remote Sensing. 2019, 11(15):1820.”

Line: 397-400: Added: “[4] Li, X. H.; Lei, S.G.; Liu, F.; Wang, W.Z. Analysis of

Plant and Soil Restoration Process and Degree of Refuse Dumps in Open-Pit Coal

Mining Areas. International Journal of Environmental Research and Public Health.

2020, 17(6): 1975.”

Line: 401-403: Added: “[5] Dominguez‐Haydar, Y.; Armbrecht, I. Response of Ants and Their Seed Removal in Rehabilitation Areas and Forests at El Cerrejón Coal Mine in Colombia. Restoration Ecology. 2011, 19: 178-184.”

Line: 404-406: Added: “[6] Luo, Z.B.; Ma, J.; Chen, F.; Li, X.X.; Zhang, Q.;Yang, Y.J. Adaptive Development of Soil Bacterial Communities to Ecological Processes Caused by Mining Activities in the Loess Plateau, China. Microorganisms. 2020, 8(4): 477.”

Line: 407-409: Added: “[7] Huang, Y.F.; Zhang, S.W.; Zhang, L.P.; Zhang, H.Y;.Li, Z. Research progress on conservation and restoration of biodiversity in land reclamation of opencast coal mine. Journal of Agricultural Machinery. 2015, 46(8): 72-82.”

Line: 410-412: Added: “[8] Ao, M.; Qiu, G.L.; Zhang C.; Xu, X.H.; Zhao, L.; Feng, X.B.; Qin, S.; Meng, B. Atmospheric deposition of antimony in a typical mercury-antimony mining area, Shaanxi Province, Southwest China. Environmental Pollution. 2018, 245: 173-182.”

Line: 413-415: Added: “[9] Zhang, Y.B and Sun, S. Study on the Reclamation and Ecological Reconstruction of Abandoned Land in Mining Area. IOP Conference Series: Earth and Environmental Science. 2020, 514(2).”

3.The presented in the introduction statistics, data – where they have been taken?

We are thankful to the reviewer’s comments. We have carefully checked and changed some sentence in the revised manuscript.

Lines 36-44 : Changed “China is the main coal energy consumer in the word and the coal mining has made great contributions to local economic development for years [1]. Seventy-five percent of the economic value of coal production worldwide comes from open pit coal mining, and in China, open pit coal mine production occupies 15% of the total coal production [2,3]. In 2006, the USA, Russia, India, China, Australia and South Africa, produced 81.9% of the total coal extracted throughout the world [4,5]. Since the industrial revolution of the 18th and 19th centuries, coal mining in the past more than 200 years has produced great contribution to the national economy, it has inevitably bring serious environmental and land problems [6,7].” to “Coal resources has always been in a very important position in the structure of nonrenewable energy in the world [1,2]. China is the main coal energy consumer in the word and the coal mining has made great contributions to local economic development for years [3]. According to the statistics, seventy-five percent of the added value of global coal production comes from open-pit coal mine, in China , open-pit coal production accounts for 15% of the total coal production [4]. At present, China has gained popularity as the world’s largest producer and consumer of coal resources [5,6]. According to forecasts, the total mineable coal reserves in China with a rate of hundreds of millions of tons per year [7].”

4.The authors presented the general data about open coal mining as for 2006, would be good to introduce more recent data. The introduction of the state with the revitalization of post-mining sites across the world is rather fragmentary. Either authors have to enrich the state with revitalization with exact cases across the world or to concentrate only at the situation in China. Currently, the global state is presented very fragmentary and does not show the real state of the art, a number of countries with intensive coal mining: GB, Czech Republic, etc. are missing.

We are thankful to the reviewer’s comments, and strongly cherish this opportunity. Therefore, we attach great importance to the above comments. We have added some informative focus on Chinese Open-Pit Coal Mines.

Lines 39-50: Changed “Seventy-five percent of the economic value of coal production worldwide comes from open pit coal mining, and in China, open pit coal mine production occupies 15% of the total coal production [2,3]. In 2006, the USA, Russia, India, China, Australia and South Africa, produced 81.9% of the total coal extracted throughout the world [4,5]. Since the industrial revolution of the 18th and 19th centuries, coal mining in the past more than 200 years has produced great contribution to the national economy, it has inevitably bring serious environmental and land problems [6,7].” to “According to the statistics, seventy-five percent of the added value of global coal production comes from open-pit coal mine, in China , open-pit coal production accounts for 15% of the total coal production [4]. At present, China has gained popularity as the world’s largest producer and consumer of coal resources [5,6]. According to forecasts, the total mineable coal reserves in China with a rate of hundreds of millions of tons per year [7]. Such as, the Shengli coalfield contains 22.4 billion tons of coal, which is the lignite coalfield with the thickest coal seam and the largest reserves in China, and is also one of the three coalfields in Inner Mongolia Autonomous Region with more than 20 billion tons, and has been included in the national large-scale coal power base (Fig 1).The large-scale extraction of coal has put country’s industry in a state of vigorous development and at the same time brings huge economic benefits”.

5.The motivation why the Shengli opencast coal mine in Xilingol League, Inner Mongolia of China was selected as one case study, is missing. Moreover, in the following investigation authors did not refer to this particular case and it became unclear whether the reviewed cases (rather often without appropriated citations) are connected with this particular site?

We appreciate the reviewer’s insightful comments. We have deleted some sentences in the revised manuscript.

Line 87-97: Deleted “The Shengli opencast coal mine in Xilingol League, Inner Mongolia of China was selected as an example to research restoration vegetation and it was an Open-Pit Coal Mine, as shown in (Fig 1). The Shengli coalfield contains 22.4 billion tons of coal, which is the lignite coalfield with the thickest coal seam and the largest reserves in China, and is also one of the three coalfields in Inner Mongolia Autonomous Region with more than 20 billion tons, and has been included in the national large-scale coal power base (Fig 2). The coal type of Shengli coal field is lignite, with 1.890 billion tons of recoverable reserves in the well field, with reliable resources and superior mining conditions. The soil type of coal mine mainly consists of chestnut calcium soil, meadow chestnut calcium soil, meadow soil, etc. Due to the degradation of the meadow, sandy and gravelly chestnut calcium soil has been formed, with low soil organic matter content and poor soil fertility (Fig 3).”

6.Figure 2 and Figure 3 are not connected with the main goal of the review which pretends to overview existing in China technologies. The pictures are not much informative, and it is not clear what was the reason to include them in the manuscript.

We appreciate the reviewer’s insightful comments. We have carefully checked and corrected this mistake in the revised manuscript.

Lines 58-59 Changed “and has been included in the national large-scale coal power base (Fig 2)” to “uneven particle size distribution, no soil aggregate structure and poor level of nutrients [22].”

Lines 61:Deleted “(Fig 3)”.

7.Rows 120-125 present common general knowledge, well known from the Class Books as per open-pit mining reclamation, but they are not informative.

We appreciate the reviewer’s insightful comments, and sorry for our negligence. We have carefully checked and corrected this mistake in the revised manuscript.

Lines 122-126: Changed “With the increase in national attention to ecological restoration in mining areas, research on soil improvement technologies in mining areas has become the focus of many researchers. Open-pit mining causes various types of soil damage [35], for example, soil depletion, topsoil stripping, and lack of biomass. With mining, the surrounding vegetation is destroyed and a large amount of soil from deep pits is deposited around the area, forming a drainage field [36]. Many factors limit plant growth in these soils, including soil pH, toxic substances in the soil, lack of nutrients, etc., creating an environment in which basically no plants can survive.” to “Many factors limit plant growth in these soils, including soil pH, toxic substances in the soil, lack of nutrients, etc., creating an environment in which basically no plants can survive.With the increase of national attention to ecological restoration in mining areas, the research of soil improvement technology in mining areas has become the focus of attention of many scholars.”

8.Table S2 of the current manuscript is very similar to Table 2 of the reference [30]: even the titles of both tables are the same and conclusions from them.  It is not explained how Table S2 was created, and to which extent it is original.

We appreciate the reviewer’s insightful comments, and sorry for our negligence. We have carefully checked and corrected this mistake in the revised manuscript.

Line 262-266: Song et al. [91] found that inoculation with AMF could significantly improve the biomass of maize and enhance the stress resistance of plants. The inoculation significantly increased the above-ground part by 28.3%, 34.8%, and 24.4%, respectively, while the underground part increased by 37.3%, 20.6%, and 34.8%, respectively (Table 2).

Line 283-287: Based on the review of studies, some scholars demonstrate that co-introduction of native plants and AMF is an effective way to establish species rich vegetation in post-mining areas, the differences in plant and AMF variables in response to addition of seeds or soil inoculum and their interaction have been summarized in Table S2[97].

9.Row 137:  ETP sludge – the full title of the meaning, not abbreviation.

We appreciate the reviewer’s insightful comments, and sorry for our negligence. We have added full title of the meaning.

Line: 137: Added: “ETP sludge (effluent treatment plant sludge) ”

10.Conclusions are rather trivial and weak: the well-known methods used for ecological restoration of post-mining sides are calculated: soil fertilization, phytoremediation, microbial remediation and their common influence. Which new knowledge has been created as a result of the study?

We appreciate the reviewer’s insightful comments. We have explained new knowledge has been created as a result of the study.

Lines 114-119: This review aimed to state the main technologies of soil improvement and function enhancement in mining areas, vegetation restoration and reconstruction, and maintaining the sustainable development of the ecosystem. We summarize the feasibility of ecological restoration technologies in mining areas from the perspective of vegetation, soil and microorganisms.

11.More comprehensive research was done before with good conclusions, like in the reference: Feng, H.B.; Zhou, J.W.; Zhou, A.G.; Bai, G.Y.; Li, Z.X.; Chen, H.N.; Su, D.H.; Han, X. Grassland Ecological Restoration Based on the Relationship between Vegetation and Its Below-ground Habitat Analysis in Steppe Coal Mine Area. Science of The Total Environment. 2021, 78: 146221.  What was the reason to present the conclusions of this reference again in the reviewed manuscript?

We are thankful to the reviewer’s comments, and strongly cherish this opportunity. Therefore, we attach great importance to the above comments.

Lines 169-172: The sentence “Through vegetation restoration and ecological rehabilitation, a stable and efficient artificial vegetation ecosystem is established in the mining area, which provides good ecological environment conditions for the survival of plants and animals [60]. ” In order to make the article more convincing, we cite this literature.

12.Row 92: The Shengli coal fields… ‘has been included in the national large-scale coal power base” however, the reference is to figure 2, not to the database.

We are thankful to the reviewer’s comments. We have carefully checked and corrected this mistake in the revised manuscript.

Lines 87-97: Deleted “The Shengli opencast coal mine in Xilingol League, Inner Mongolia of China was selected as an example to research restoration vegetation and it was an Open-Pit Coal Mine, as shown in (Fig 1). The Shengli coalfield contains 22.4 billion tons of coal, which is the lignite coalfield with the thickest coal seam and the largest reserves in China, and is also one of the three coalfields in Inner Mongolia Autonomous Region with more than 20 billion tons, and has been included in the national large-scale coal power base (Fig 2).

13.Rows 133-134. It is written ‘some researchers in India”: references are missing.

We appreciate the reviewer’s insightful comments, and sorry for our negligence. We have added some references.

Lines 139: Added “[30] Jambhulkar H P and Hemlata P. Eco-restoration approach for mine spoil overburden dump through biotechnological route. Environmental monitoring and assessment. 2019, 191(12): 1-16.” 

14.Rows 193-194. It is written “Based on a review of published studies, to achieve vegetation restoration in mining areas, the plant type has an important impact on the effect of vegetation restoration; thus, the ratio of different plant species should be considered”: references are missing.

We appreciate the reviewer’s insightful comments, and sorry for our negligence. We have added some references.

Lines 201: Added “[65]Zhang L, Zhaohua L U, Tang S, et al. Slope vegetation characteristics and community stability at different restoration years of open-pit coal mine waste dump. Acta Ecologica Sinica, 2021, 41(14):5764-5774. and [66]Zhang S, Mi J, Hou H, Yang Y. Research progress of mine ecological restoration – based on the report of three consecutive world ecological restoration conferences. Acta Ecol. Sin, 2018, 38, 5611-5619.” 

15.Rows 213-215. It is written “Based on our previous experience in mine restoration, we concluded that the mixed grass-bush-tree or bush-tree mode attains the best effect in regulating soil bulk density”: references are missing.

We appreciate the reviewer’s insightful comments, and sorry for our negligence. We have added some references.

Lines 214-215: Added “[68] Zhao Y, Chai L J, Chen J, et al. Technology and appli-cation for ecological rehabilitation on self-maintaining veg-etation restoratio. Land Reclamation Ecological Fragile Areas, 2017,10: 255-257. and [69] Li Q S, Han X, Zhao Y, et al. Research on integration and application of key technologies of Ecological restoration and Management of vegetation restoration in open-pit coal mine - A case study of the outfall of Shengli open-pit Mine. Environmental Ecology, 2021, 3(6):7.”

16.Rows 277-278. It is written “In addition, some studies demonstrated that co-introduction of native plants and AMF is an effective approach to establish species-rich vegetation in post-mining areas”: references are missing.

We appreciate the reviewer’s insightful comments, and sorry for our negligence. We have added some references.

Lines 287: Added “[97] Vahter T, Bueno C G, Davison J, et al. Co-introduction of native mycorrhizal fungi and plant seeds accelerates restoration of post‐mining landscapes. Journal of Applied Ecology. 2020, 57(9): 1741-1751.”

17.Rows 309-313. It is written “Currently, technologies for post-mine ecological restoration monitoring in China are still in the development stage. The most common method of environmental monitoring is field surveys and satellite remote sensing, which are relatively primitive. These methods are not only inefficient and costly but the timeliness and accuracy of the obtained data need to be improved”: references are missing.

We appreciate the reviewer’s insightful comments, and sorry for our negligence. We have added some references.

Lines316-318: Added “[102]Shuai S, Zhang Z, Lyu X, et al. Remote sensing monitoring of vegetation phenological characteristics and vegetation health status in mine restoration areas. Nongye Gongcheng Xuebao/Transactions of the Chinese Society of Agricultural Engineering, 2021, 37(4):224-234. and [103]Qi Y N, Liao S B, Wang Q L. Research on evaluation of mining area ecological security based on GF-1 satellite imagery-taking Fushun West open-pit mine for example. IOP Conference Series: Earth and Environmental Science, 2021, 783(1):012128 (7pp).”

18.Rows 315-317. It is written, “In Table 3, we summarize and compare the advantages and disadvantages of manual surveying, satellite, remote sensing, and UAV remote sensing data collection based on previous research”: which research? references are missing.

We appreciate the reviewer’s insightful comments, and sorry for our negligence. We have added some references.

Lines316-318: Added “[104]Rasmussen J, Azim S, Jensen S M, et al. The challenge of reproducing remote sensing data from satellites and unmanned aerial vehicles (UAVs) in the context of management zones and precision agriculture. Precision Agriculture, 2020. and [105]Liao X, Yue H, Liu R, et al. Launching an unmanned aerial vehicle remote sensing data carrier:concept,key components and prospects. International Journal of Digital Earth, 2020(010):013.”

Thank you again for your comments. We hope we could learn more from you. Finally, we are very grateful to the reviewer’s and editor’s understanding and affirmation again. We wish the article can be published in Agriculture.

Reviewer 4 Report

An interesting work, but the approach was not what I expected when I saw the title of the work. In this

L1: Type of article?

in certain areas of the introduction only a bibliographic source is given. only a bibliographic source cannot support "more research" as it is written in the paper. And in some areas the bibliography is missing, being very important because it supports the statements of certain studies/researchers (For examples L 47, L 121, L 179)

What I noticed in the paper is the fact that certain information is given only partially. This applies to each section. I was expecting concrete examples of plants, microorganisms, etc. that were used and obtained x, y, z yields

  L 77: "Many effective studies...", but there are only 3 studies. possibly to be added: "Sustainable Ecological Restoration of Sterile Dumps Using Robinia pseudoacacia, https://doi.org/10.3390/su132414021, "Preliminary Investigations Regarding the Potential of Robinia pseudoacacia L. (Leguminosae) in the Phytoremediation of Sterile Dumps. Journal of Environmental Protection and Ecology 21, No 1, 46-55 (2020)

I think that after L 86, paragraph L 106-116 should be introduced, slightly modified so that the introduction is separated from the activity carried out within this revision.

  Fig 5: I don't see its importance in the work. What is its purpose? I think it is sufficient to explain it in the text, if you really want an image/diagram, I recommend that you establish exactly how to place the plants. For example, no tree is presented here, but in table 1 it is mentioned....something is not connected. The authors give the impression that the article is still a work in progress, being in fact an outline of their work

L 208: 12 site? where from? these sites are not specified in the work

L 243: Here I recommend the use of the following source and not only: "Extraction of Metals from Polluted Soils by Bioleaching in Relation to Environmental Risk Assessment". https://doi.org/10.3390/ma15113973

L 268: examples of biological fertilizers. It must be specified what exactly is used

Table 1 needs improvements in terms of appearance

Table 1: here the term "tree" is mentioned for the first time...it is not clear what is with it, why trees have not been discussed until now.

 Fig 7. where did these actions take place? did this studio make it? what conclusions were drawn in this study? The bibliographic source is missing

  The word "thesis" is used a lot in the work. Why? I propose to use a single term to avoid confusion

Only after major revisions, this paper can be considered for publication

Author Response

Revision Notes

Dear Editor:

Thank you for your kind letter of “Agriculture-2126182 - Editor decision -Major Revisions” on December 28, 2022.

Generally, we appreciate the editor and reviewer’s insightful comments, which are helpful for improving the manuscript. We are thankful to reviewer #1、#2、#3 and #4 comments , which are all valuable and very helpful for revising and improving our paper. Based on editor and reviewers' comments and requests, we have made moderate modification in the revised manuscript. Below is a summary of our responds to the reviewers' comments.

In revision notes, the line numbers refer to the PDF vision of the revised manuscript.

We have highlighted the changes in blue according to reviewer #4 comments in the annotated version of the revised manuscript (Revision, changes marked).

1.L1: Type of article? in certain areas of the introduction only a bibliographic source is given. only a bibliographic source cannot support "more research" as it is written in the paper. And in some areas the bibliography is missing, being very important because it supports the statements of certain studies/researchers (For examples L 47, L 121, L 179).

We appreciate the reviewer’s insightful comments. we have added some informative and sentences in the revised manuscript.

Lines 100-101: Added “[41] Sur I M. Sustainable Ecological Restoration of Sterile Dumps Using Robinia pseudoacacia. Sustainability, 2021, 13. and [42]Babau A, Micle V, Damian G, et al. Preliminary Investigations Regarding the Potential of Robinia pseudoacacia L. (Leguminosae) in the Phytoremediation of Sterile Dumps. Journal of environmental protection and ecology, 2020,(1):21.

Lines 122: Added “[51] Nadalia D and Pulunggono H B. Soil characteristics of post-mining reclamation land and natural soil without top soil. Journal of Degraded and Mining Lands Management, 2020(2).”

Lines 214-215: Added “[68]Zhao Y,Chai L J,Chen J, et al. Technology and appli-cation for ecological rehabilitation on self-maintaining veg-etation restoratio. Land Reclamation Ecological Fragile Areas, 2017,10: 255-257. and [69]Li Q S, Han X, Zhao Y, et al. Research on integration and application of key technologies of Ecological restoration and Management of vegetation restoration in open-pit coal mine - A case study of the outfall of Shengli open-pit Mine. Environmental Ecology, 2021, 3(6):7.”

Line 243: Added “ [78] Sur I M, Micle V, Hegyi A, et al. Extraction of Metals from Polluted Soils by Bioleaching in Relation to Environmental Risk Assessment. Materials, 2022,15(11):393.”

2.What I noticed in the paper is the fact that certain information is given only partially. This applies to each section. I was expecting concrete examples of plants, microorganisms, etc. that were used and obtained x, y, z yields.

We appreciate the reviewer’s insightful comments, and sorry for our negligence. We have added some informative about yields.

Lines 151-153: Added “ The results showed that the optimal formula of substitute material was a ratio of m (subclay):m (peat) = 20:1 and an applied microbial agent concentration of 0.15 kg/m2 [57].”

Lines 154-157: Added “ Lu et al. [58] found that optimum proportion was found to be soil: UCFA: MSLs = 70:20:10 and soil: UCFA: MSLs = 60:20:20, Which achieved excellent results in Baorixile Open-Pit mine remediation applications in Inner Mongolia, China .”

Lines 256-258: Added “ Bi et al. [87] found that inoculation with AM fungi increased the seed yield and aboveground biomass of wheat by 46.6% and 56.5%, respectively;”

Lines 2626-266: Added “ Song et al. [91] found that inoculation with AMF could significantly improve the biomass of maize and enhance the stress resistance of plants. The inoculation significantly increased the above-ground part by 28.3%, 34.8%, and 24.4%, respectively, while the underground part increased by 37.3%, 20.6%, and 34.8%, respectively (Table 2)”

3."Many effective studies...", but there are only 3 studies. possibly to be added: "Sustainable Ecological Restoration of Sterile Dumps Using Robinia pseudoacacia, https://doi.org/10.3390/su132414021, "Preliminary Investigations Regarding the Potential of Robinia pseudoacacia L. (Leguminosae) in the Phytoremediation of Sterile Dumps. Journal of Environmental Protection and Ecology 21, No 1, 46-55 (2020).

We appreciate the reviewer’s insightful comments, and sorry for our negligence. We have added some references .

Lines 101: Added references“ [41] Sur I M. Sustainable Ecological Restoration of Sterile Dumps Using Robinia pseudoacacia. Sustainability, 2021, 13. and [42]Babau A, Micle V, Damian G, et al. Preliminary Investigations Regarding the Potential of Robinia pseudoacacia L. (Leguminosae) in the Phytoremediation of Sterile Dumps. Journal of environmental protection and ecology, 2020,(1):21.”

4.I think that after L 86, paragraph L 106-116 should be introduced, slightly modified so that the introduction is separated from the activity carried out within this revision.

We appreciate the reviewer’s insightful comments, and sorry for our negligence. Considering the overall structure of the article, we have deleted this part.

Lines 87-97: Deleted “The Shengli opencast coal mine in Xilingol League, Inner Mongolia of China was selected as an example to research restoration vegetation and it was an Open-Pit Coal Mine, as shown in (Fig 1). The Shengli coalfield contains 22.4 billion tons of coal, which is the lignite coalfield with the thickest coal seam and the largest reserves in China, and is also one of the three coalfields in Inner Mongolia Autonomous Region with more than 20 billion tons, and has been included in the national large-scale coal power base (Fig 2). The coal type of Shengli coal field is lignite, with 1.890 billion tons of recoverable reserves in the well field, with reliable resources and superior mining conditions. The soil type of coal mine mainly consists of chestnut calcium soil, meadow chestnut calcium soil, meadow soil, etc. Due to the degradation of the meadow, sandy and gravelly chestnut calcium soil has been formed, with low soil organic matter content and poor soil fertility (Fig 3).”

5.Fig 5: I don't see its importance in the work. What is its purpose? I think it is sufficient to explain it in the text, if you really want an image/diagram, I recommend that you establish exactly how to place the plants. For example, no tree is presented here, but in table 1 it is mentioned....something is not connected. The authors give the impression that the article is still a work in progress, being in fact an outline of their work.

We are thankful to the reviewer’s comments. We have carefully checked and corrected this mistake in the revised manuscript.

Lines 727-728: Changed “Table 1. Restoration abundance threshold and plant allocation ratio at different site conditions [53].” to “Table 1. Restoration abundance threshold and plant allocation ratio at different site conditions [67].”

Lines 212-215: Changed “ Based on our previous experience in mine restoration, we concluded that the mixed grass-bush-tree or bush-tree mode attains the best effect in regulating soil bulk density (Fig 5).” to “According to the previous experience of mine restoration, we concluded that in the process of land reclamation and ecological restoration in mining areas, for slopes with slope ratio greater than 1:1.7, 3S-OER slope vegetation ecological protection was constructed by combination of shrub and grass[68], and the mixed grass-bush-tree or bush-tree mode attained the best effect in regulating soil bulk density[69] (Fig 4).”

6.L 208: 12 site? where from? these sites are not specified in the work.

We are thankful to the reviewer’s comments. We have carefully checked and corrected this mistake in the revised manuscript.

Line 204: Deleted “ Liu et al.[29] investigated the calculated proportions of different plant distributions in the 12 site conditions in the semi-arid mining areas of western China, with higher proportions of herbs and shrubs than trees. Herbaceous plants and shrubs play an important role in the revegetation of semi-arid mining areas (Fig. 6). ”

7.L 243: Here I recommend the use of the following source and not only: "Extraction of Metals from Polluted Soils by Bioleaching in Relation to Environmental Risk Assessment". https://doi.org/10.3390/ma15113973.

We are thankful to the reviewer’s comments. We have added literature in the revised manuscript.

Line 243: Added “ [78] Sur I M, Micle V, Hegyi A, et al. Extraction of Metals from Polluted Soils by Bioleaching in Relation to Environmental Risk Assessment. Materials, 2022,15(11):393.”

8.L 268: examples of biological fertilizers. It must be specified what exactly is used.

We are very sorry for not clearly describing this content in the previous manuscript. We have added some informative in the revised manuscript.

Lines 272-273: Added “Previous studies reported that inoculation of biological fertilizers (Glomus and Gigaspora were separately propagated in a green house in polypropylene) and application of amendments can reduce the toxicity of heavy metals in mines, eg, chromium, zinc, and cadmium, were significantly reduced to 41%, 43%, and 40%, respectively [94].”

9.Table 1 needs improvements in terms of appearance.

We are thankful to the reviewer’s comments. We have carefully checked and corrected this mistake in the revised manuscript.

Lines 727-728: We have immprovements in terms of appearance of Table 1.

10.Table 1: here the term "tree" is mentioned for the first time...it is not clear what is with it, why trees have not been discussed until now.

We are thankful to the reviewer’s comments. We have carefully checked and corrected this mistake in the revised manuscript.

Lines 727-728: Changed “Table 1. Restoration abundance threshold and plant allocation ratio at different site conditions [53].” to “Table 1. Restoration abundance threshold and plant allocation ratio at different site conditions [67].”

11.Fig 7. where did these actions take place? did this studio make it? what conclusions were drawn in this study? The bibliographic source is missing.

We appreciate the reviewer’s insightful comments. We have added some informative about Fig 7 in the revised manuscript.

Lines 311-313: Added “ For example, we conducted relevant experiments in the open-pit dump of Shengli Coal Mine in Inner Mongolia, where the slope was covered with straw, which significantly increased the seed germination rate (Fig. 5).”

12.The word "thesis" is used a lot in the work. Why? I propose to use a single term to avoid confusion.

We are thankful to the reviewer’s comments. We have carefully checked and corrected this mistake in the revised manuscript.

Lines 342: Changed “thesis” to “paper”.

Thank you again for your comments. We hope we could learn more from you. Finally, we are very grateful to the reviewer’s and editor’s understanding and affirmation again. We wish the article can be published in Agriculture.

Round 2

Reviewer 2 Report

Authors referred to all my comments. The work in its current form is consistent with its title, the descriptions used are correct and the literature references are appropriate. The work can be published in its current form.

Author Response

Revision Notes

Dear Editor:

Thank you for your kind letter of “Agriculture-2126182 - Editor decision -Minor Revisions” on January 10, 2023.

Generally, we appreciate the editor and reviewer’s insightful comments, which are helpful for improving the manuscript. We have checked all references are relevant

to the contents of the manuscript. We are thankful to reviewer #1、#2、#3 and #4 comments , which are all valuable and very helpful for revising and improving our paper. Based on editor and reviewers' comments and requests, we have made moderate modification in the revised manuscript. Below is a summary of our responds to the reviewers' comments.

In revision notes, the line numbers refer to the PDF vision of the revised manuscript.

We have highlighted the changes in red according to reviewer #2 comments in the annotated version of the revised manuscript (Revision, changes marked).

1.Authors referred to all my comments. The work in its current form is consistent with its title, the descriptions used are correct and the literature references are appropriate.The work can be published in its current form.

We are thankful to the reviewer’s comments, and strongly cherish this opportunity. Therefore, We have carefully checked the revised manuscript.

Thank you again for your comments. We hope we could learn more from you. Finally, we are very grateful to the reviewer’s and editor’s understanding and affirmation again. We wish the article can be published in Agriculture.

Reviewer 3 Report

The manuscript was essentially improved and additional literature sources were utilized. However, there is still some work to be done explained below. 

 1.       No answer for suggestion #2, i.e.: The authors did not explain which methodology they utilized. How did they provide the selection of the articles? Did they overview any scientific database like Web of Knowledge, Scopus, etc.? Or they just analyzed the known before publications and their own research?

2.       No reaction for suggestion #6 related to figure 2: it is now connected with the following explanation (lines 56-58) “After coal mining, the refuse dumps soil is mostly deep below one hundred meters, and overburdens are devoid of soil characteristics [21], uneven particle size distribution, no soil aggregate structure and poor level of nutrients [22] (Fig. 2). It is not clear how the establishments related “poor nutrients, uneven particle size distribution, etc.” are connected with the figure 2?   

3.       No reaction to suggestion #12, what does “National large-scale coal power base” mean? Any citation?

4.       Table S2 was taken from source [97] and modified, it has to be noticed in the title of the Table.

5.       Essential improvement of English style is recommended, using synonyms. During the correction of the initial version of the manuscript, the text was changed that sometimes led to duplications, for example, lines 31-33 and lines 37-39 are almost the same, etc.  The overall layout of the sentences  has to be improved using English grammar style.    

Author Response

Revision Notes

Dear Editor:

Thank you for your kind letter of “Agriculture-2126182 - Editor decision -Major Revisions” on December 28, 2022.

Generally, we appreciate the editor and reviewer’s insightful comments, which are helpful for improving the manuscript. We have checked all references are relevant

to the contents of the manuscript. We are thankful to reviewer  #1、#2、#3 and #4 comments , which are all valuable and very helpful for revising and improving our paper. Based on editor and reviewers' comments and requests, we have made moderate modification in the revised manuscript. Below is a summary of our responds to the reviewers' comments.

In revision notes, the line numbers refer to the PDF vision of the revised manuscript.

We have highlighted the changes in yellow according to reviewer #3 comments in the annotated version of the revised manuscript (Revision, changes marked).

1.No answer for suggestion #2, i.e.: The authors did not explain which methodology they utilized. How did they provide the selection of the articles? Did they overview any scientific database like Web of Knowledge, Scopus, etc.? Or they just analyzed the known before publications and their own research?

We are thankful to the reviewer’s comments, and strongly cherish this opportunity. We are very sorry for not clearly describing this content in the previous manuscript.

Lines 19-23: The methodology we summarized the integrated technology of open pit mine reclamation with microbial restoration technology as the core, ecological vegetation restoration as the essential, and soil restoration and improvement as the promotion.

We refer to one hundred literature, combined with the reality of mine rehabilitation engineering technology summary and provide the selection of the articles.

We refer to scientific database like Wed of science, Cnki and Wangfang data, etc, some articles from the last ten to twenty years.

2.No reaction for suggestion #6 related to figure 2: it is now connected with the following explanation (lines 56-58) “After coal mining, the refuse dumps soil is mostly deep below one hundred meters, and overburdens are devoid of soil characteristics [21], uneven particle size distribution, no soil aggregate structure and poor level of nutrients [22] (Fig. 2). It is not clear how the establishments related “poor nutrients, uneven particle size distribution, etc.” are connected with the figure 2?

We appreciate the reviewer’s insightful comments. We are very sorry for not clearly describing this content in the previous manuscript.

About reaction for suggestion #6 related to figure 2. The soil type of coal mine is mainly composed of chestnut soil, meadow chestnut soil, meadow soil, etc. Due to the degradation of grassland, sandy and gravel chestnut soil has been formed, with low soil organic matter content and poor soil fertility. Figure 2 shows the dump made from the soil dug up after mining in the real engineering open-pit. After testing, it was found that the soil nutrients were poor and they were all waste soil with large particle sizes.

3.No reaction to suggestion #12, what does “National large-scale coal power base” mean? Any citation?

We appreciate the reviewer’s insightful comments. We are very sorry for not clearly describing this content in the previous manuscript.

About the reaction to suggestion #12, according to the opinions of other reviewers, the introduction of Inner Mongolia Nortel Shengli Coal Mine is not very meaningful, and the content of the whole paper is not highly consistent. In the end, we decided deleted the section.

Line 87-97: Deleted “The Shengli opencast coal mine in Xilingol League, Inner Mongolia of China was selected as an example to research restoration vegetation and it was an Open-Pit Coal Mine, as shown in (Fig 1). The Shengli coalfield contains 22.4 billion tons of coal, which is the lignite coalfield with the thickest coal seam and the largest reserves in China, and is also one of the three coalfields in Inner Mongolia Autonomous Region with more than 20 billion tons, and has been included in the national large-scale coal power base (Fig 2). The coal type of Shengli coal field is lignite, with 1.890 billion tons of recoverable reserves in the well field, with reliable resources and superior mining conditions. The soil type of coal mine mainly consists of chestnut calcium soil, meadow chestnut calcium soil, meadow soil, etc. Due to the degradation of the meadow, sandy and gravelly chestnut calcium soil has been formed, with low soil organic matter content and poor soil fertility (Fig 3).”

4.Table S2 was taken from source [97] and modified, it has to be noticed in the title of the Table.

We are thankful to the reviewer’s comments, and strongly cherish this opportunity. We are very sorry for not clearly describing this content in the previous manuscript.

We have noticed in the title of the Table in supplementary materials.

Lines 43-49 (Supplementary materials): Table S2 Differences in plant and arbuscular mycorrhizal fungal (AMF)variables in response to the addition of seeds or soil inoculum and their interaction. Degrees of freedom (df), F-statistics and p values are reported. Significance at 0.001***, 0.01** and 0.05* levels shown. Ns- not significant. The results of generalized linear models with the addition of seeds and soil inoculum as fixed effects are shown [97]. Notes: Degrees of freedom (df), F-statistics and p values are reported. Significance at 0.001***, 0.01** and 0.05* levels shown. Ns- not significant.

  1. Essential improvement of English style is recommended, using synonyms. During the correction of the initial version of the manuscript, the text was changed that sometimes led to duplications, for example, lines 31-33 and lines 37-39 are almost the same, etc. The overall layout of the sentences has to be improved using English grammar style.

We are thankful to the reviewer’s comments. We have carefully checked and corrected this mistake in the revised manuscript. We also asked the relevant English professional organization (Enago) staff to polish the language. The overall layout of the sentences has been improved using English grammar style.

Lines 37-41:Changed “Seventy-five percent of the economic value of coal production worldwide comes from open pit coal mining, and in China, open pit coal mine production occupies 15% of the total coal production [2,3].” to “China is the main coal energy consumer in the word and the coal mining has made great contributions to local economic development for years [3]. According to the statistics, seventy-five percent of the added value of global coal production comes from open-pit coal mine, in China , open-pit coal production accounts for 15% of the total coal production [4].”

Thank you again for your comments. We hope we could learn more from you. Finally, we are very grateful to the reviewer’s and editor’s understanding and affirmation again. We wish the article can be published in Agriculture.

Reviewer 4 Report

it's good! I recommend publication in the Agriculture

Author Response

Revision Notes

Dear Editor:

Thank you for your kind letter of “Agriculture-2126182 - Editor decision -Minor Revisions” on January 10, 2023.

Generally, we appreciate the editor and reviewer’s insightful comments, which are helpful for improving the manuscript. We have checked all references are relevant

to the contents of the manuscript. We are thankful to reviewer #1、#2、#3 and #4 comments , which are all valuable and very helpful for revising and improving our paper. Based on editor and reviewers' comments and requests, we have made moderate modification in the revised manuscript. Below is a summary of our responds to the reviewers' comments.

In revision notes, the line numbers refer to the PDF vision of the revised manuscript.

We have highlighted the changes in blue according to reviewer #4 comments in the annotated version of the revised manuscript (Revision, changes marked).

1.it's good! I recommend publication in the Agriculture.

We are thankful to the reviewer’s comments, and strongly cherish this opportunity. Therefore, We have carefully checked the revised manuscript.

Thank you again for your comments. We hope we could learn more from you. Finally, we are very grateful to the reviewer’s and editor’s understanding and affirmation again. We wish the article can be published in Agriculture.
